# Salicylic Acid’s Impact on Growth, Photosynthesis, and Antioxidant Enzyme Activity of *Triticum aestivum* When Exposed to Salt

**DOI:** 10.3390/molecules28010100

**Published:** 2022-12-23

**Authors:** Pravej Alam, Thamer Al Balawi, Mohammad Faizan

**Affiliations:** 1Department of Biology, College of Science and Humanities in Al-Kharj, Prince Sattam bin Abdulaziz University, Al-Kharj 11942, Saudi Arabia; 2Botany Section, School of Sciences, Maulana Azad National Urdu University, Hyderabad 500032, India

**Keywords:** abiotic stress, salicylic acid, biochemical pathway, transpiration rate

## Abstract

Recently, the application of salicylic acid (SA) for improving a plant’s resistance to abiotic stresses has increased. A large part of the irrigated land (2.1% out of 19.5%) is severely affected by salinity stress worldwide. In 2020, total production of wheat (*Triticum aestivum*) was 761 million tons, representing the second most produced cereal after maize; therefore, research on its salinity tolerance is of world concern. Photosynthetic attributes such as net photosynthetic rate (P_N_), stomatal conductance (gs), intercellular CO_2_ concentration (Ci), and transpiration rate (E) were increased significantly by the application of SA. Salt stress increased antioxidant enzyme activity; however, SA further boosted their activity along with proline level. We conclude that SA interacts with meristematic cells, thereby triggering biochemical pathways conductive to the increment in morphological parameters. Further research is required to dissect the mechanisms of SA within the wheat plants under stress.

## 1. Introduction

Salt stress is one of the major constraints for the environment; it reduces plant growth, mainly in arid and semi-arid conditions [1]. Soil salinity affects almost 800 million hectares of land all over the world [2]. Salt stress is primarily detected by the root system, and it alters plant metabolism by activating osmotic stress due to less water availability and ion toxicity because of nutrient imbalance [3,4]. The toxic impacts of salt stress can differ on the basis of climatic conditions, intensity of light, plant type, and soil profile [5]. Salinity stress also manifests as oxidative stress guided by ROS. As a result, ion toxicity directly leads to chlorosis and necrosis, mostly due to Na^+^ accretion that obstructs with several physiological practices in plants [6]. All these responses to salt stress have injurious effects on plants [7]. To mitigate the toxicity caused by salt stress, various molecules have been used. In the present study, SA was used as an anti-toxicity agent.

Plants are composed of various growth regulators commonly known as phytohormones, which contribute to diverse plant activities, pathways, and regulating mechanisms at minimum concentration. Some of the significant growth regulators are gibberellic acid (GA), ethylene, auxin, cytokinins (CKs), salicylic acid, and brassinosteroids (BRs) [8]. These hormones work in extreme abiotic stress conditions such as salinity, drought, heat stress, waterlogging stress, heavy metal, and cold stress. Among them, salicylic acid (SA) is a phenolic compound consisting of various antioxidant substances. SA comes from the scientific name of the plant willow tree (*Salix alba*), and it was first extracted from bark of the tree in 1928. At first, its name was salicin, but it later changed to SA for its oxidation capability. It is also known as ortho-hydroxybenzoic acid, which is a colorless, complex, crystalline compound that helps to produce aspirin. Various metabolic pathways, such as flowering and synthesis, can be used to isolate SA in free or conjugated form in the environment [9,10]. It widely contributes to plant growth, development, respiration, conductance, and assimilation, especially in stress management. SA actively participates in stomatal conductance, nutrient elevation, and mobility of enzymes [11]. Plant modulation in stress is strengthened by the application of SA in oxidative stress. Physiological, morphological, and biochemical metabolism is altered through the use of SA during environmental stress in plants. Germination rate, transpiration, and defense mechanisms are also influenced by SA. It acts as signaling molecule and activates enzymatic functions to trigger abiotic stressors. The stress-responsive marker gene is associated with SA application under challenging ecological conditions [12]. Genes such as *TLC1* are induced in in vivo conditions, are activated transcriptionally, and promote signaling [13].

Exogenous SA application helps in improving antioxidant efficacy in various biological schemes [14]. SA plays an indispensable role in reactive oxygen species (ROS) regulation, e.g., hydrogen peroxide. Antioxidant enzyme (SOD, CAT, POD, and GR) regulation in oxidative stress is increased by SA induction. In *Haematococcus pluvialis*, exogenous SA induces the ROS activities of SO, APX, and CAT [15]. Antioxidant upregulation is also found in wheat, pepper, and mustard. Biosynthetic enzyme regulators and proteins are also induced. Phenylalanine acts as aromatic amino-acid precursor which leads to SA synthesis [16]. Positive SA applications have been found in soybean, maize, rice, and sunflower [17]. Cell-responsive and metabolic activities are found in various protein components of *Cucumis sativus* which are SA-induced [18]. In maize, the photosynthetic rate and carbohydrate metabolism are increased during salinity stress [19]. PSII activity is controlled by SA application in wheat due to the upregulation of antioxidant activities [20]. ATP sulfurylase, enzymatic, and NR activity in mungbean is also stimulated through SA application [21]. The enzymatic action of ascorbate–glutathione pathway synthase maintains the redox balance of a plant [22]. SA decreases the proline contents in leaves and stabilizes membrane activities. Proline accumulation causes deleterious effects in plant metabolism, which can be mitigated by exogenous SA application [23]. K^+^ leakage in root tissues is lessened and H^+^-ATPase activity is enhanced in *Arabidopsis* plants, thereby increasing the plasma membrane exchange capacity and cytosol accumulation [24]. *NahG* transgenic *Arabidopsis* lines are facilitated by the application of SA [25]. Rubisco activity and pigmentation biosynthesis are stimulated by SA usage [22].

Physiological parameters of plants such as relative water content (RWC) are also controlled by SA. In plants, an increase in RWC contributes to adaptation in adverse climatic conditions, sustainability, and water balance [26]. SA treatment works to counterbalance chlorophyll content, and the chl a/b ratio varies in different stress conditions. SA represents the genotype-dependent marker responses of chlorophyll [27]. Pretreatment with SA maintains quick leaf senescence and prevents oxidation damage in plants, which led to improved RWC levels in barley [28]. Lower SA concentration contributes to enhanced wheat seed pigmentation and a reduction in transpiration rate, thus evoking stomatal closure [29]. The uptake of micro- and macro-mineral components from underground is mitigated in saline conditions through foliar SA application. The interactive role in mineral and nutrient uptake still needs to be explored. Synergistic and antagonistic crosstalk between plant hormones plays a vital role in abiotic stress management. ABA and IAA accumulation occurred due to SA application in wheat and corn [30]. Cd stress is alleviated by SA application with an auxin-mediated responsive state, and SA is sensitized in the indole-3-acetic acid (IAA) pathway in the roots of maize [31]. An antagonistic effect is regulated between SA and jasmonic acid (JA) signaling by the mitogen-activated protein kinase (MAPK) signaling pathway [32]. Wheat (*Triticum aestivum*) is the most important staple food crop produced all over the world, native to Southeast Asia and widely cultivated since prehistoric times in temperate zones. Wheat not only is an important source of starch and energy in the diet, but also provides a substantial amount of various essential and beneficial components [33].

From the published literature, it is hypothesized that SA can also be used as a micronutrient to enhance plant growth and development under normal and saline conditions. Considering the above-described data, an experiment was conducted to dissect the impact of SA on the improvement of the morpho-physiology of the wheat plants. Seeds were primed with SA (500 µM) before sowing, whereas NaCl (150 mM) was given through soil.

## 2. Results

### 2.1. Effect of SA on Growth Attributes under NaCl Stress

The growth parameters revealed that application of SA through seed soaking significantly increased all growth indices in wheat plants compared to control (Figure 1A–F). It was observed that NaCl drastically reduced plant growth and development when applied via soil. However, SA significantly reduced the toxicity caused by NaCl in wheat plants. It is clearly displayed in the Figure 1A–F that plants whose seeds were soaked in SA prior to sowing and NaCl after sowing led to renewed growth and development of wheat plants (Figure 1A–F).

### 2.2. Photosynthesis and Related Attributes Influenced by NaCl

Figure 2A–E demonstrate the toxicity of wheat plants exposed to NaCl. The effects of SA are also shown in the same figure, illustrating the positive effects on the chlorophyll index and photosynthesis attributes. The phytotoxicity of wheat caused by NaCl was also reduced by SA (Figure 2A–E) following completion of the experiment.

### 2.3. Activity of Antioxidant Enzymes in the Presence of SA and/or NaCl

Antioxidant enzyme activity plays an important role in the plant defense mechanism. In this experiment, SOD, CAT, and POX activities and proline content were significantly increased in the plants that received NaCl. However, these activities were further augmented with SA, as shown in Figure 3A–D.

## 3. Discussion

Salinity stress is a severe serious abiotic stress affecting plant productivity worldwide. The Food and Agricultural Organization (FAO) published a report describing the annual agricultural loss, which is approximately 20–40% due to salinity stress [34]. At the global level, salt stress negatively influences crop growth, disrupts the cellular, metabolic, and physiological performance, and ultimately reduces the growth of developing crops [35]. Salinity stress may disturb the Calvin cycle, photosystems, stomatal functioning, and electron transport chain [36]. To overcome the toxicity generated by salt stress, SA is a very good molecule, because SA is an endogenously secreted signaling molecule that triggers plant defense mechanisms against stresses. It plays a significant role in regulating abiotic stress tolerance via thermoregulation, protects against oxidative stress, and influences different phases of the plant life cycle [37]. SA results in the accumulation of osmolytes, disturbs mineral nutrition uptake, enhances the scavenging power of ROS, boosts the deposition of secondary metabolites, and initiates the biosynthesis pathways of other plant hormones [38]. It was confirmed from the present experiment that NaCl severely reduced wheat growth; however, SA increased plant growth in the presence/absence of salt stress in comparison with control plants (Figure 1). The reason behind the growth enhancement by SA is that it can regulate growth by altering cell division and expansion. Investigations revealed that NahG transgenic plants showed higher expression of the cell-cycle G1/S transition regulator cyclin D (CYC3) and increased endoreduplication concentrations, which led to larger cells [39]. Another way to increase plant growth is through the accumulation of tryptophan biosynthesis, which is responsible for a speedy growth rate. This evidence confirms that SA plays an important role in increasing the growth of wheat plants in the presence/absence of NaCl, which is in accordance with a previous study on maize after SA treatment [40].

During salinity stress, the leaf chlorophyll index and plant photosynthetic rate are significantly decreased [41]. The leaf is the primary site of photosynthesis, and the accumulation of chlorophyll is directly related to the performance of photosynthesis [42]. Results of the present study revealed that NaCl significantly degraded chlorophyll in leaves, while SA (seed soaking) significantly alleviated the phytotoxicity caused by NaCl in terms of the chlorophyll index, as shown in Figure 2. Basirat and Mousavi [43] reported that SA recovered the chlorophyll content under high-temperature stress in Cucumis sativus. Salinity stress can also reduce P_N_ and the synthesis of organic matter, thus influencing plant growth [44]. In Gossypium, salinity toxicity considerably reduced growth, weight, photosynthesis, and related attributes [45]. Results of this study concluded that NaCl decreased P_N_, gs, Ci, and E while SA increased these attributes in the presence/absence of salt stress (Figure 2). These results are in line with previous studies in Zea mays under salt stress [19] and Triticum aestivum under drought stress [46].

One of the damaging factors of salinity stress is the induction of ROS production, such as superoxide (O_2_^−^) and hydrogen peroxide (H_2_O_2_) [47]. Antioxidant enzymes protect the cell structure against ROS formation under stress conditions [48]. Salt stress tolerance can be certified to increase antioxidant enzyme activity, thus reducing oxidative damage in plant cells. In the present study, under salinity stress conditions, a considerable increase in the activities of CAT, POX, and SOD was observed in response to oxidative damage. ROS production is an important mechanism to maintain the tolerance of plants under stress conditions. CAT and SOD are defense enzymes that scavenge O_2_^−^ radicals into H_2_O_2_, which is further detoxified to water [49]. The increased activities of antioxidative enzymes under salinity stress indicate that CAT, POX, and SOD play an important role in scavenging superoxide radicals during salt stress. Application of SA further boosted the activity of CAT, POX, and SOD, as displayed in Figure 3A–C. Therefore, the cumulative impact of CAT, POX, and SOD plays a prime role in the detoxification of ROS in plants, thus minimizing cellular injury due to ROS under salinity stress. The application of SA in salt-stressed plants induced CAT, POX, and SOD activity in the leaves. SA increases the activity of antioxidant enzymes, decreases ROS content, and consequently reduces oxidative damage to the membranes [21,50]. The stimulatory effects of SA on SOD, POX, and CAT performance were previously confirmed by various researchers in pistachio [51,52]. In line with the previous reports, the present study showed that SA can play an important role in modulating CAT, POX, and SOD activity in wheat under salinity stress (Figure 3A–C). Along with antioxidant enzyme activity, the proline content also increased with salt stress, and the level further increased upon the application of SA (Figure 3D). Previous studies also revealed that salinity impacts the proline content in Triticum aestivum [53], Hordeum vulgare [23], Torreya grandia [54], and soybean [55]. In plants, proline may also help to regulate leaf water potential (LWP) under salt stress [56].

## 4. Materials and Methods

### 4.1. Growth Conditions and Treatments

*Triticum aestivum* seeds were surface-sterilized with sodium hypochlorite for 5 min and then washed with double-distilled water (DDW). The sterilized seeds were sown in pots, which were filled with soil and manure, and then allowed to grow under natural environmental conditions with photosynthetically active radiation (PAR) of 960 µmol/m^2^/s. Prior to sowing, seeds were soaked in 500 µM SA for 12 h. At 15 days after sowing (DAS), NaCl (150 mM) was administered to the seedlings through the soil. The treatments of this experiment were as follows: control, SA (500 µM), NaCl (150 mM), and NaCl (150 mM) + SA (500 µM). A simple randomized block design was followed, and different parameters were studied at the stage of 30 days.

### 4.2. Growth Parameters

Plant growth was measured in the form of length and weight. Shoot and root length were measured using a meter scale, whereas fresh weight was calculated using a weighing machine and dry weight was recorded after drying in an oven at 70 °C for 72 h.

### 4.3. Chlorophyll Index

The chlorophyll index was calculated using a SPAD chlorophyll meter (SPAD-502; Konica, Minolta Sensing, Inc., Sakai, Osaka, Japan).

### 4.4. Photosynthesis and Related Attributes

The net photosynthetic rate (P_N_), stomatal conductance (gs), intercellular CO_2_ concentration (Ci), and transpiration rate (E) of the plant were measured using a portable infrared gas analyzer (LiCOR 6200, Portable Photosynthesis System, Lincoln, NA, USA).

### 4.5. Antioxidant Enzymes

For the estimation of antioxidant enzymes, the leaf tissue (0.5 g) was homogenized in 50 mM phosphate buffer (pH 7.0) containing 1% polyvinylpyrrolidone. The mixture was centrifuged at 15,000× *g* for 10 min at 4 °C, and the resulting supernatant was used as a source for estimating the enzyme activities of catalase (CAT, 1.11.1.6), peroxidase (POD, EC 1.11.1.7), and superoxide dismutase (SOD, EC 1.15.1.1). For the estimation of POX activity, the enzyme extract (0.1 mL) was added to a reaction mixture of pyrogallol, phosphate buffer (pH 6.8), and 1% H_2_O_2_. The change in absorbance was read every 20 s for 2 min at 420 nm [57]. A control mixture was prepared by adding double-distilled water (DDW) instead of enzyme extract. The reaction mixture for CAT consisted of phosphate buffer (pH 6.8), 0.1 M H_2_O_2_, and enzyme extract (0.10 mL). Sulfuric acid (H_2_SO_4_) was added to the reaction mixture, and, after its incubation for 1 min at 25 °C, it was titrated against potassium permanganate solution (KMnO_4_) [57]. The activity of SOD was assayed by measuring its ability to inhibit the photochemical reduction of nitroblue tetrazolium (NBT) following the method of Beauchamp and Fridovich [58]. The reaction mixture consisted of 50 mM phosphate buffer (pH 7.8), 20 µM riboflavin, 75 mM NBT, 13 mM methionine, and 0.1 mM ethylenediaminetetraacetic acid (EDTA). The mixture was illuminated with two fluorescent light tubes (40 µmol·m^−2^·s^−1^) for 10 min. The absorbance was measured at 560 nm using a UV–visible spectrophotometer.

The method of Bates et al. [59] was used for the identification of proline content in young leaves. Leaves were extracted in sulfosalicylic acid, and equal volumes of glacial acetic acid and ninhydrin solution were also added. The sample was heated at 100 °C, and then 5 mL of toluene was added. Absorbance of the aspired layer was read at 528 nm on a spectrophotometer. The proline content was expressed as µg·g^−1^ FW.

### 4.6. Statistical Analysis

SPSS was used to conduct the analysis of variance (ANOVA). A significant difference was considered at *p* < 0.05.

## 5. Conclusions

From the present study, it can be concluded that NaCl significantly reduced wheat growth and photosynthesis, along with the chlorophyll index. However, SA treatment of the seeds through soaking overcame the toxicity caused by NaCl. Proline content and antioxidant enzymes also played an important role in minimizing the deleterious effects of ROS within the plant cell. Salinity stress increased the activities of CAT, POX, and SOD, which were further augmented upon the application of SA. In the future, the exact mechanism of action of SA under salinity stress will be investigated.

## Figures and Tables

**Figure 1 molecules-28-00100-f001:**
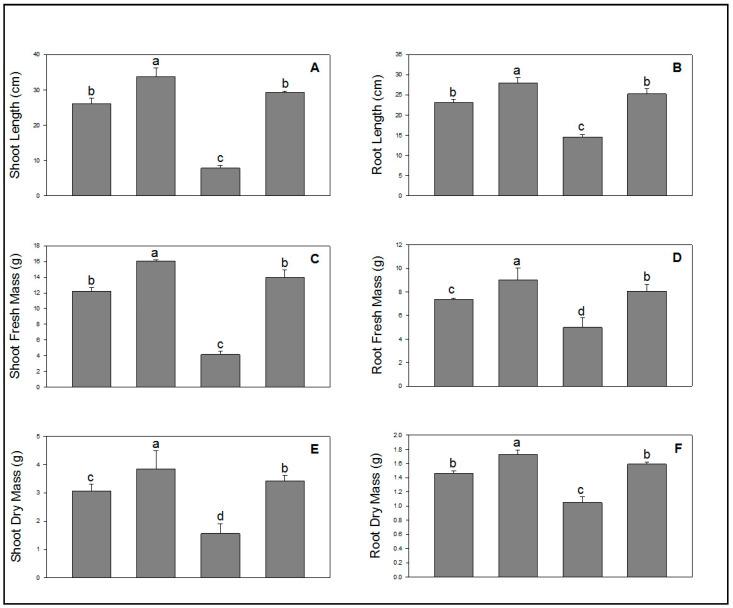
Effect of salicylic acid (500 µM) on shoot length (**A**), root length (**B**), shoot fresh mass (**C**), root fresh mass (**D**), shoot dry mass (**E**), and root dry mass (**F**) of wheat at 30 DAS under NaCl (150 mM) stress. All data are the mean of five replicates (*n* = 5), and vertical bars shows standard errors (±SE).

**Figure 2 molecules-28-00100-f002:**
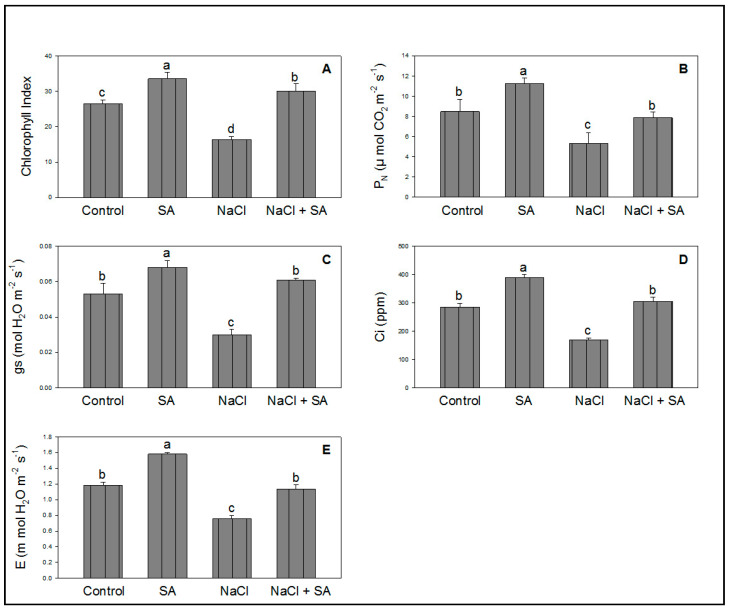
Effect of salicylic acid (500 µM) on chlorophyll index (**A**), P_N_ (**B**), gs (**C**), Ci (**D**), and E (**E**) of wheat at 30 DAS under NaCl (150 mM) stress. All data are the mean of five replicates (*n* = 5), and vertical bars shows standard errors (±SE).

**Figure 3 molecules-28-00100-f003:**
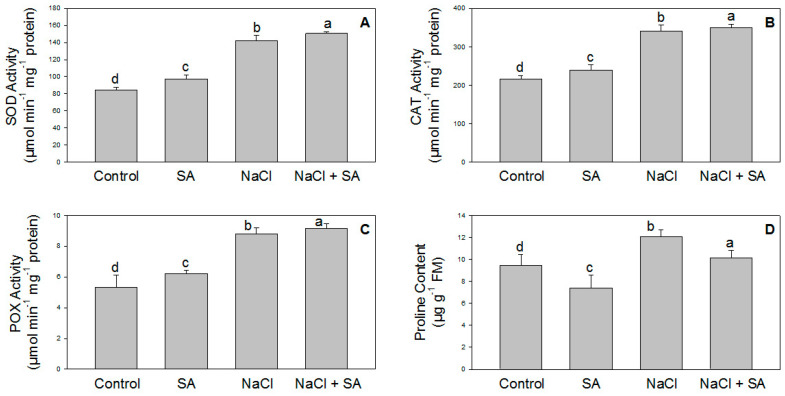
Effect of salicylic acid (500 µM) on the activity of SOD (**A**), CAT (**B**), and POX (**C**), as well as on the proline content (**D**), of wheat at 30 DAS under NaCl (150 mM) stress. All data are the mean of five replicates (*n* = 5), and vertical bars shows standard errors (±SE).

## Data Availability

Not applicable.

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
