# Peer review of "Salicylic Acid’s Impact on Growth, Photosynthesis, and Antioxidant Enzyme Activity of Triticum aestivum When Exposed to Salt"

_molecules, 2022, doi:10.3390/molecules28010100_

Round 1
Reviewer 1 Report
In this manuscript by Alam et al., the authors found in wheat, pre-treatment with salicylic acid (SA) can improve its salt tolerance. It indicated an potential application of SA for abiotic stress.
Minor comments:
- In the introduction, "SA decreases proline contents in leaves and stabilizes membrane activities. " (Line 77), while in this study, the authors found more proline accumulates after application of SA. It needs to be discussed.
- Line 81, the citation is incorrect.
- Line 82, "Rubisco activity along with pigmentation biosynthesis is stimulated by SA usage" needs citation.
- Lines 105-107 should be deleted.
- Line 135, "sown" should be "shown"
Author Response
Author responses to reviewer (1) comments
(Molecules-2055696 R1)
To Reviewer #1
Remarks: In this manuscript by Alam et al., the authors found in wheat, pre-treatment with salicylic acid (SA) can improve its salt tolerance. It indicated an potential application of SA for abiotic stress.
Response: The authors are very thankful to the anonymous Reviewer for the appreciation, valuable suggestions, comments and scientific criticism of manuscript for its further improvement.
Remarks: In the introduction, "SA decreases proline contents in leaves and stabilizes membrane activities." (Line 77), while in this study, the authors found more proline accumulates after application of SA. It needs to be discussed.
Response: Corrected, data puts wrongly by mistake in the figure.
Now, Corrected.
Remarks: Line 81, the citation is incorrect.
Response: Corrected.
Response: Line 82, "Rubisco activity along with pigmentation biosynthesis is stimulated by SA usage" needs citation.
Response: Added.
Remarks: Lines 105-107 should be deleted.
Response: Corrected.
Remarks: Line 135, "sown" should be "shown"
Response: Corrected.
The authors are very thankful to the anonymous #Reviewer1 for the appreciation, valuable suggestions, comments and scientific criticism of manuscript for its further improvement. All the suggestions and comments of the have been accepted by the authors and the manuscript has been corrected accordingly. A thorough internal reviews was also performed in the whole MS for possible improvement, changes highlighted in Track Change Format supplied MS. We hope the response meets the reviewer approval.

Reviewer 2 Report
The Authors show interesting data, but the manuscript required an improvement. They focus on the salicylic acid effect on some parameters of Triticum aestivum. Nevertheless, in the introduction any sentence introduces the crop under study. I suggest to include some comments on it:
- Pandino et al., 2020. Organic cropping system affects grain chemical composition, rheological and agronomic performance of durum wheat. Agriculture 10 (2), 46
- Scavo et al., 2022. Allelopathy in Durum Wheat Landraces as Affected by Genotype and Plant Part. Plants 11 (8), 1021
L15-19 : This section should be delete
L105-107: Should be delete
Materials and Methods
- Growth conditions and treatments
This section required a major revision. Where the experiment was perfomed (open field or under controlled conditions)? Which are the climatic or controlled condition? Chararacteristic of soil? How the NaCl was applied?. As reported by Authors in L33-34, these informations are very important to define. In addition, how many replicate were perfomed?
4.3 - 4.4 - 4.5
Only references are reported, a brief description for each parametes should be included. The Authors report some data on proline, but here is missing.
Author Response
Author responses to reviewer (2) comments
(Molecules-2055696 R1)
To Reviewer #2
Remarks: The Authors show interesting data, but the manuscript required an improvement. They focus on the salicylic acid effect on some parameters of Triticum aestivum. Nevertheless, in the introduction any sentence introduces the crop under study. I suggest to include some comments on it:
Response: Corrected.
Remarks: L15-19: This section should be delete.
Response: Corrected.
Remarks: L105-107: Should be delete.
Response: Corrected.
Remarks: Materials and Methods
- Growth conditions and treatments
This section required a major revision. Where the experiment was performed (open field or under controlled conditions)? Which are the climatic or controlled conditions? Chararacteristic of soil? How the NaCl was applied?. As reported by Authors in L33-34, these information’s are very important to define. In addition, how many replicate were perfomed?
Response: Corrected.
Remarks: 4.3 - 4.4 - 4.5
Only references are reported, a brief description for each parameters should be included. The Authors report some data on proline, but here is missing.
Response: Corrected.
The authors are very thankful to the anonymous #Reviewer2 for the appreciation, valuable suggestions, comments and scientific criticism of manuscript for its further improvement. All the suggestions and comments of the have been accepted by the authors and the manuscript has been corrected accordingly. A thorough internal reviews was also performed in the whole MS for possible improvement, changes highlighted in Track Change Format supplied MS. We hope the response meets the reviewer approval.

Round 2
Reviewer 2 Report
The Authors improved their work following the suggested revisions